# Living Together Apart: Quantitative Perspectives on the Costs and Benefits of a Multipartite Genome Organization in Viruses

**DOI:** 10.3390/v17091275

**Published:** 2025-09-20

**Authors:** Marcelle L. Johnson, Dieke Boezen, Alexey A. Grum-Grzhimaylo, René A. A. van der Vlugt, J. Arjan G. M. de Visser, Mark P. Zwart

**Affiliations:** 1Department of Microbial Ecology, Netherlands Institute of Ecology (NIOO-KNAW), 6700 AB Wageningen, The Netherlandsa.grum@wi.knaw.nl (A.A.G.-G.); 2Laboratory of Virology, Wageningen University and Research, 6700 AA Wageningen, The Netherlands; rene.vandervlugt@wur.nl; 3Food and Indoor Mycology, Westerdijk Fungal Biodiversity Institute, Uppsalalaan 8, 3584 CT Utrecht, The Netherlands; 4Laboratory of Genetics, Wageningen University and Research, 6700 AA Wageningen, The Netherlands; arjan.devisser@wur.nl

**Keywords:** multipartite virus, segmented virus, plant virus, evolution, dose response, infection model, gene-product sharing, host range

## Abstract

Background: Multipartite viruses individually package their multiple genome segments into virus particles, necessitating the transmission of multiple virus particles for effective viral spread. This dependence poses a cost in the form of reduced transmission compared to monopartite viruses, which only have a single genome segment. The notable cost of a multipartite genome organization has spurred debate on why multipartite viruses are so common among plant viruses, including a search for benefits associated with this organizational form. Methods: We investigated the costs and benefits of multipartite viruses with three approaches. First, we reanalyzed dose–response data to measure the cost of multipartition to between-host transmission for multipartite viruses. Second, we developed a simulation model to explore when the sharing of viral gene products between cells is beneficial. Third, we tested whether multipartite viruses have a broad host range by estimating the host range for plant viruses using metagenomics data. Results: We find that the observed cost to transmission exceeds theoretical predictions. We predict that a virus gene-product-sharing strategy only confers benefits under limited conditions, suggesting that this strategy may not be common. Our results suggest that multipartite and segmented viruses have broader host ranges than monopartite viruses. Conclusions: Our analyses also suggest there is limited evidence for the costs and benefits of a multipartite organization, and we argue that the diversity of multipartite virus–host systems demands pluralistic explanatory frameworks.

## 1. Introduction

Virus genomes are organized into single or multiple nucleic acid molecules, which we refer to as genome segments. These genome segments are packaged into virus particles, facilitating spread of the infection to other cells, organs or hosts. Viruses with single genome segments are termed monopartite viruses. Viruses with multiple genome segments differ in their strategies for packaging these segments into virus particles. Viruses which package all segments into a single virus particle are called segmented viruses, whereas viruses that package each segment into a separate virus particle are called multipartite viruses [1]. For segmented viruses, the actual distribution of genome segments over virus particles shows considerable diversity: virus particles with missing genome segments can be common [2], or even predominate in the non-selectively packaging segmented viruses [3]. Both segmented and multipartite viruses require the presence of multiple genome segments containing core viral functions for the initiation of viral replication and subsequent propagation in hosts. Multipartite viruses are always multicomponent systems, as are segmented viruses with appreciable numbers of incomplete particles, because in both cases, transmission depends on multiple virus particles. This raises the possibility that successful transmission may not occur because core genome segments are lost, imposing a cost on transmission compared to viruses which produce only complete virus particles (i.e., the monopartite viruses and selectively packaging segmented viruses) [1,4,5,6,7,8].

Mathematical models of the relationship between virus dose and the prevalence of infection in hosts, henceforth referred to as “dose response”, suggest this cost to transmission is appreciable [4,6,7,8,9]. Despite having an organization that is not conducive to virus spread, multipartite viruses are common [1]. These viruses represent ~40% of plant virus genera [10], are common among the fungal viruses [8] and there is growing evidence of at least some multipartite viruses that infect animal hosts [8,11,12]. A key question which has fueled debate since the discovery of multipartite viruses is, therefore, why viruses with low expected transmissibility are so common [1]. Whereas we focus mainly on multipartite viruses in this reflection, much of the reasoning presented is also relevant to segmented viruses.

Historically, three main lines of reasoning have been proposed to address the intriguing question of why multipartite viruses are so common. First, it has been postulated that there are benefits to a multipartite organization that outweigh its cost. For example, re-assortment between segments could bring together high-fitness segments [13,14], and shorter segments could fit within smaller, potentially more stable virus particles [15] or resolve problems related to the fidelity of translation [16]. A recent suggestion is that rapid changes in the frequency of virus genome segments (the genome formula) could be adaptive because these changes may tune gene expression in different host environments [9,17,18]. However, for the plant viruses, there is not yet convincing empirical support for these ideas [1]. Furthermore, many of the proposed benefits, including the evolutionary benefits of re-assortment and genome-formula change, only require genome segmentation and, therefore, could pertain to both multipartite and segmented viruses [1,2,19]. Second, it has been proposed that multipartition may not be as costly as suggested by theory. The cost could be reduced if virus particles aggregate [20] or if genome segments form supramolecular structures (when within-host spread can be achieved by unencapsidated segments) [21], although there is no convincing evidence for either proposal yet. It has been shown that viral gene products are present in cells where the corresponding genome segment is absent, potentially facilitating within-host spread by allowing distributed replication [22]. A recent study demonstrates that incomplete multipartite virus infections can re-acquire segments [23]. Although the incomplete virus populations used in these experiments contained all segments required for replication, these observations suggest that non-concomitant transmission of core segments may reduce the cost to transmission. Third, it has been proposed that a multipartite organization emerges because of cheating at the within-host level [14,24]. In this context, cooperators are full-length genomes encoding all gene products required for replication, whereas cheaters are defective viral genomes that no longer make all gene products but outcompete their full-length counterparts in coinfection. Theory suggests that cheaters can displace co-operators under a wide range of conditions, and that by implication multipartition could emerge whilst not having any benefits on higher levels of selection (i.e., at the between-host level) [25]. In summary, there is a lively and unresolved debate on the reasons for the emergence of multipartite viruses, inspired by both experimental and theoretical work.

As there are a number of excellent reviews on the emergence and evolution of multipartite viruses [1,8,10], here, we will not give a comprehensive overview of the field. Instead, we will reflect on three key issues to help frame the debate on why multipartite viruses are common, using a quantitative, data-driven perspective. First, we will consider the cost of a multipartite organization to between-host transmission, an important issue because assumptions about cost underpin many of the explanations for multipartition. The mechanism that underlies this cost is clear: one or more essential segments may not enter the new host [5,6,9,20]. But how strong is the empirical evidence for this cost of multipartition, and can this cost be quantified? Second, multiple mechanisms that decrease the cost to multipartite virus transmission have been proposed [20,22,23]. Here, we develop a simple simulation model to consider the costs and benefits of disseminating viral gene products over host cells [22], to illustrate the complexities associated with mechanisms that are assumed to reduce transmission costs. Third, re-assortment and genome-formula change will allow for considerable genome variation in multipartite and segmented viruses, which may allow for a broader host range. We perform a preliminary test of this hypothesis by estimating virus host ranges for monopartite, segmented and multipartite plant viruses, using an untargeted viral detection approach based on publicly available high-throughput sequencing (HTS) data.

## 2. Results and Discussion

### 2.1. The Transmission Cost of Multipartition

Multipartite plant viruses were discovered because of their aberrant dose–response relationships, a discovery that occurred due to the unique virus–host interactions that occur in these systems. In some host plants, viruses may induce local lesions, i.e., readily visible necrotic responses on the inoculated leaves, which can be quantified by counting. By preparing a dilution series of a viral inoculum, challenging plants with these various viral doses, recording the responses and then comparing these data to model predictions, it is possible to infer single-hit infection kinetics for many monopartite plant viruses [26]. However, for some viruses, the rate of infection decreased more steeply than predicted by single-hit models [26,27], suggesting a multicomponent system. More definitive results were obtained by considering the dose response of the tri-segmented multipartite prunus necrotic ringspot virus (PNRSV). For this virus, direct evidence was found for genetic complementation between virus particles [4]. Fulton showed that partially UV-inactivated PNRSV particles increased the infectivity of the inoculum disproportionately. From these strong synergistic interactions between the viral genetic components, he inferred that: “Two or more virus particles at one site might provide a complete complement of genetic units […]” [4]. Although Fulton demonstrated synergism between virus particles, the implication of these observations is also that—all other things being equal—a multipartite virus has impaired transmission compared to a monopartite virus, an effect that is strongly dose-dependent. Subsequent research has provided further support for this conclusion. It has generally been observed that monopartite viruses have dose–response relationships that match single-hit predictions of the independent action hypothesis (IAH) model, whereas multipartite viruses show a steeper dose response [6,28,29,30].

Here, we contend that these analyses alone are insufficient to establish that the cost of multipartition is real and matches theoretical predictions. Classical studies on local lesions consider the relationship between the relative dose (inferred from the dilution) and the number of local lesions, in a dose range where the number of local lesions is not close to saturation [4,26]. This approach also works for other infection readouts, such as primary infection foci [30] and plaque formation in cell culture [12]. The gradient of the dose–response curve is independent of the virus dose used and the infectivity of the virus, i.e., the probability of infection per virus particle [31]. As a result, classic experiments could be performed without knowledge of the absolute virus dose (i.e., only knowing relative doses), and the gradients could be compared for different viruses without needing to account for different levels of infectivity. Similarly, more recent work has considered host infection as a binary response variable, resulting in a sigmoidal dose response [31]. In these cases, the shape of the dose response is determined by the number of virus particle types required for infection, whereas the position of the curve has not been considered experimentally [6,29]. Theory predicts that dose response will become steeper as the number of virus particle types required for infection is increased, but equally important, the position of the dose–response curve shifts to the right in a predictable manner as it becomes less likely that the virus will successfully infect (Figure 1a) [6,7]. To date, the shifts in the position of the dose–response curve have not been included in analyses of virus infections. The combined effect of the shifted dose response and its altered shape ultimately determine the cost of multipartition to transmission. Therefore, dose–response analyses to date evaluate whether the experimental results are compatible with a given number of segments, but they do not evaluate whether the cost of multipartition is in agreement with model predictions.

A comprehensive comparison of the dose response, considering both shape and position for natural monopartite and multipartite viruses to quantify the transmission cost, will be confounded by genetic differences between viruses and their effects on infectivity of viruses. In other words, we do not have viruses that differ only in the number of genome segments at our disposition for such a test. Therefore, a more satisfactory approach is to engineer a panel of hosts with different requirements for infection, for example by expressing viral genome segments so that they are no longer needed for infection. The same inoculum could then be used to challenge the different hosts, but different segment numbers would be required for infection depending on the host being challenged. In this case, we could compare the positions of the dose–response curve and evaluate whether they coincide with theoretical predictions, as is the case for the shape of the curve. This would shed light on whether the cost to infectivity of multipartition matches model predictions. In previous work, the dose response was measured for tobacco plants constitutively expressing one or two genome segments of the tripartite alfalfa mosaic virus (AMV) [32], but the authors only considered the shape of the dose response [6]. Here, we have re-analysed these data to consider both the shape and position of the response, fitting a range of different dose–response models.

Dose–response experiments were performed with a wildtype AMV inoculum in three tobacco plants [6]: wildtype plants (Wt), plants expressing AMV RNA2 (P2) and plants expressing both AMV RNA1 and RNA2 (P12) [32]. A simple model of infection that assumes (i) independent entry into host cells by the three RNA segments during viral invasion, and (ii) that all three segments need to be present for a productive infection was used to generate predictions (see Methods Section 4.1 for a complete description). We tested a set of four models in which the probability of infection per virus particle type (i.e., containing a particular genome segment) was fixed (Model 1), or alternatively, it was dependent on the genome segment (Model 2), the host plant type (Model 3), or both on genome segment and host plant (Model 4) (Figure 2).

We fitted these models with a maximum likelihood approach and performed model selection with the Akaike information criterion (AIC), which considers both the fit and complexity (i.e., number of free model parameters) of a model. For a detailed overview of the results, see Appendix A. The Akaike weight (AW) indicates the relative likelihood of a model [33]. We obtained the following AW values: Model 1: 0, Model 2: 0.328, Model 3: 0.632, and Model 4: 0.040. Model 4 is the best-fit model, but its complexity is higher than Models 2 and 3, resulting in a low AW value (see also Table A2 in Appendix A). We can therefore reject Models 1 and 4. Models 2 and 3 both have considerably more support, although we cannot make a meaningful distinction between these two models in terms of empirical support. When we plot the predicted dose response, it is clear that Model 1 predicts that dose–response curves corresponding to the different experimental conditions will be closer together than observed from the data (Figure 1b). Models 2–4 better fit the data and predict larger differences in the dose responses over the different types of host plants (Figure 1c). In sum, we found that the cost of multipartition was larger than predicted by the null model of infection (Model 1, Figure 1b). However, support for two alternative models is similar, and therefore, we cannot identify which of these two mechanisms might underlie this larger-than-expected cost.

Our analysis provides support for the idea that multipartition has a considerable cost to transmission, although we cannot clearly identify why this cost is higher than predicted. However, our results and previous analyses provide some clues. First, under Model 2 the estimated probability of infection is appreciably higher for RNA3 than for the other two RNAs (Figure A1 in Appendix A). Under model 3, the probability of infection is higher in the plants expressing both AMV RNA1 and RNA2, than in any other plants (Figure A1). These results suggest that infection kinetics are markedly different in the plants expressing two AMV segments. In previous work, the AMV genome formula was measured upon infection of the transgenic plants used here [18]. The AMV genome formula was similar for wildtype plants and those expressing one AMV segment, but shifted towards very high levels of RNA3 for plants that expressed RNA1 and RNA2. In other work, primary infection loci also appeared to be largest in the plants expressing AMV RNA1 and RNA2 [6]. Taken together, these results suggest that the plants expressing RNA1 and RNA2 have become highly susceptible to AMV, removing barriers to primary and systemic infection.

For the experimental data used, plants were exposed to AMV by mechanical inoculation. Although mechanical inoculation is a relevant transmission route for many plant viruses, for most plant viruses, vector-borne transmission is the foremost route [34,35,36]. Vector probing may introduce virus particles to a very specific location, rather than the leaf surface, as during mechanical inoculation. The transmission route may have a major effect on the cost of multipartition. The cost of multipartition to vector-borne transmission also may vary enormously depending on many factors, including vector density, vector behavior and other vector-host interactions, and the time of transit between hosts [37,38]. For example, ecological settings that generate many opportunities for vector-borne transmission may alleviate this cost [39]. Tests of the cost to vector-borne transmission would therefore be valuable, but they will need to be performed over a variety of conditions, virus–host systems, and potentially different vector types to be informative.

Here we reanalyzed data obtained for AMV, a member of the *Bromoviridae* family, for which all members have tri-segmented genomes. We speculate that the cost to transmission will be similar to model predictions for other multipartite RNA and DNA viruses, for measurements performed under similar conditions (i.e., mechanical inoculation). However, there are some important caveats to consider. First, whereas AMV requires all segments for replication [18], some viruses have segments with accessory functions [40], limiting the number of segments required to establish infection and possibly lowering the cost to transmission. The loss of segments with accessory functions may very well incur other costs, if they are not recuperated [23] later on. Second, as the number of segments increases, the marginal cost of each additional segment will be lower, all other things being equal [7]. The small differences in cost when the segment number is higher than three may be difficult to measure experimentally. This is illustrated by the partial overlap in the confidence band for *P2* and wild-type plants, requiring two and three segments for infection, respectively, that we observed here (Figure 1b,c). Third, the frequency of segments will impact infection kinetics: As a segment becomes rarer, it will be a limiting factor in establishing infection. Paradoxically, rare segments will therefore further increase the cost to transmission, while also resulting in a dose-relationship with a similar shape to a monopartite virus, as only one segment in effect limits the proportion of infected hosts [6]. Similar patterns apply if virus particles encapsidating different segments have different probabilities of entering the host (e.g., Model 2). Therefore, although we expect similar model predictions for the cost to transmission for other multipartite viruses, the characteristics of each system are crucial and need to be considered carefully.

### 2.2. Reducing Multipartite Virus Transmission Costs by Gene-Product Sharing

Several putative mechanisms that reduce the cost of multipartite transmission have been proposed [20,21,22,23]. One proposal with considerable empirical evidence is the sharing of viral gene products between cells, as it has convincingly been shown for faba bean necrotic stunt virus (FBNSV) that viral proteins can be present in cells in which the cognate DNA segment is missing [22]. The ability to share gene products across cells and tissues circumvents the requirement for all segments to be present, thereby reducing the cost to within-host spread of a multipartite virus. This phenomenon has been demonstrated for FBNSV gene products involved in key viral functions: replication, movement and coat protein synthesis [22]. Although gene-product sharing appears to be a plausible manner to reduce the cost of multipartite virus spread within the host, sharing gene products will have an inherent cost: some resources in an infected “donor cell” will be directed to the expression of gene products in “recipient cells”. This cost to the donor cell will depend on two factors. First, the currently unknown mechanism of gene-product sharing between donor and recipient cells will determine the cost. Trafficking viral proteins will likely come at a higher cost to the donor cell than the movement of mRNA between cells, as it will limit the resources for protein translation in the donor cell. Second, the intensity of gene-product sharing will also determine its cost to the donor cell. Both are currently unknown, and the cost of viral gene-product sharing is therefore unclear. Moreover, the benefits will also depend on the likelihood that shared gene products augment cellular infection, which will depend on factors such as the cellular multiplicity of infection and the distribution of shared gene products over cells. As these factors are currently unknown, we developed a model of multipartite virus infection to explore under which conditions gene-product sharing may be beneficial, focusing on a minimal model to aid its interpretation.

We extended a simulation model [9] of a bipartite virus incorporating variation in the genome formula (Figure 3a,b). In the original model, the intra-cellular genome formula and virus particle yield per cell are linked by the probability density function of the normal distribution, with mean *µ* and variance *σ*^2^. Therefore, parameter *σ*^2^ determines the magnitude of the decrease in virus particle yield as the genome formula departs from its optimal value *µ*, meaning that *σ*^2^ determines how sensitive virus particle yield is to changes in the genome formula. In the original study, it was shown that multipartite viruses can outcompete their monopartite cognates when virus replication is sensitive to changes in gene expression (i.e., when *σ*^2^ ≤ 0.1).

Here, we extended this model to include the production of both virus gene products and virus particles in cells. The model is illustrated in Figure 3a,b and described in detail in Methods Section 4.2. Here, we only highlight three key features of the model we developed. First, the frequency of the two gene products of the bipartite virus (i.e., one gene product per genome segment) in a cell will determine the total amount of gene products synthesized in that cell, whereas the frequency of genome segments will determine which gene products are made. Second, a fraction *ρ* of gene products synthesized within each infected cell is shared with other cells and is not available for virus particle production in that cell. The sharing of these gene products results in a proportional decrease in virus particles produced in the donor cell. Third, the model assumes discreet rounds of infection, in which a fixed number of susceptible cells is exposed to the virus particles produced in the previous round of infection. Shared gene products are distributed homogeneously over the susceptible cells introduced for each round of infection.

From our model results (Figure 3c,d), we can draw three main conclusions. First, for a large part of the parameter space we considered, our model predicts that viral gene-product sharing is neutral or costly. As the fraction of gene products shared (*ρ*) becomes larger, the fitness cost becomes higher. However, at very low multiplicities of cellular infection (MOI), the cost becomes negligible as the multipartite virus performs very poorly. If gene products are targeted to cells that are likely to be exposed to virus particles, this could increase virus benefit, but the model incorporates no such mechanism. Second, gene-product sharing was beneficial in a small region of parameter space: moderate levels of sharing (*ρ* < 0.5), low MOI (~3), and when virus replication is insensitive to the genome formula (*σ*^2^ = 10). Intuitively, there are benefits associated with gene-product sharing under these conditions. Moderate levels of sharing reduce the opportunity cost to replication in the donor cell. At low MOI, gene-product sharing will be beneficial because a single genome segment will be present in many cells. Low sensitivity to the genome formula allows virus replication in cells in which only a small amount of a missing segment has been shared. Third, the benefits of rapid genome-formula change are predicted to outweigh the cost of multipartition when virus particle production is sensitive to genome-formula change (*σ*^2^ ≤ 0.1) [9]. The benefits of gene-product sharing occur in a different parameter space than the benefits associated with adaptive genome-formula change, making it unlikely that both mechanisms are beneficial at the same time. Therefore, the model result suggests the two explanations are mutually exclusive. For a more extensive exploration of model results over a broader range of conditions, see Appendix B (Appendix B and Figure A3).

Our results suggest that although gene-product sharing has the potential to reduce the cost of multipartition at the within-host level, there are many conditions under which it imposes an additional cost. Given the specific set of conditions needed, we predict that the occurrence of gene-product sharing to lower the cost of multipartition may not be common, unless (1) there are mechanisms that target virus particles and gene products to susceptible cells, or (2) the cost to virus particle production is lower than assumed in the model. Although the model we present is agnostic about the mechanism of sharing, it assumes gene product levels produced are determined solely by the resources available in the donor cell. This assumption better fits protein trafficking than movement of mRNAs, as the latter exploits the translation machinery and amino acid pool of the recipient cell. We assumed uniform diffusion of shared gene products from infected to susceptible cells. Future work could consider the role of spatial structure, in particular the dissemination of virus particles and shared gene products to susceptible cells.

FBNSV is a phloem-limited virus, which means it replicates in a relatively small number of highly interconnected cells, perhaps facilitating the dissemination of virus gene products to cells that are likely to be exposed to virus particles. Moreover, the high size exclusion limit between sieve elements and companion cells [41] may contribute to enabling easier dissemination of virus gene products for phloem-limited viruses, compared to viruses that also replicate in other tissues.

To estimate the fitness of a virus, we considered the total virus particle production in the population of susceptible cells over time. We used a different metric than in previous work [9], in which we considered the outcome of head-to-head competition. We do not use this approach here, as gene-product sharing is inherently vulnerable to exploitation when virus variants are in direct competition. The interactions between virus variants with different levels of gene-product sharing could be highly complex, and perhaps add further restrictions on when sharing might be favorable. However, our approach is not suitable for addressing this question, as we focused on a simple model to determine under which conditions sharing confers direct benefits to a single virus variant. To explore when gene-product sharing is an evolutionary stable strategy (ESS) would require a model that explores infection dynamics for multiple virus variants within a host. To properly balance the costs and benefits of gene-product sharing in a mixed-variant environment will require more complex models that incorporate the spatial structure of plant cells, mechanisms of coinfection exclusion, and continuous viral infection.

Finally, we note that gene-product sharing may also result in a cost to between-host transmission. When replication without a complete genome is enabled by gene-product sharing, genome segments will be missing in the virus particles produced in these cells. Therefore, virus particles from these cells can only contribute to between-host transmission when they are complemented by the missing segments, during virus acquisition or upon concomitant transmission of virus populations.

### 2.3. Benefits of Genome Segmentation: Evidence for Extended Host Range?

One possible benefit of genome segmentation is increased genome plasticity, which will be enabled by different mechanisms. First, re-assortment occurs in both segmented and multipartite viruses and has potential evolutionary benefits, including host range expansion as a consequence of new combinations of genome segments [14,42,43,44,45]. Second, rapid changes in the genome formula may also be adaptive, for example, by tuning or stabilizing gene expression in different host environments [1,9,17,46]. The genome formula is unbalanced for all multipartite viruses in which it has been measured [11,17,18,47,48,49], as well as for some segmented viruses [19,50,51]. Putative benefits related to genome-formula changes may only be accessible to a subset of segmented viruses, in particular those with flexible packaging of genome segments into virus particles (e.g., the non-selectively packaging segmented viruses). Third, in some cases genome segmentation may bypass constraints on genome length, enabling the acquisition of additional functions. For example, nidovirus genome size is thought to be constrained by translation fidelity, and genome segmentation was therefore necessary to acquire novel functions [16].

If multipartite viruses and a subset of segmented viruses have greater genome plasticity, this may confer virus populations with the ability to adapt rapidly to novel environments. This rapid adaptability may affect processes at the within-host level, including the invasion of different host tissues and the evasion of host immunity. It may also affect between-host processes, allowing viruses to adapt to novel host species, thereby increasing their host range. While a broad host range is not necessarily adaptive, it can be advantageous under certain ecological circumstances, such as when host species fluctuate or if spillover into new hosts affords additional opportunities for transmission. Thus, segmentation and multipartition need not be adaptive because they expand host range. Rather, greater genome plasticity may enhance viral adaptability across different ecological contexts, with broader host range as one possible outcome that can be measured. Therefore, we predict that the host range of multipartite and segmented viruses is broader than that of monopartite viruses. Moreover, because genome-formula changes may not be pervasive for segmented viruses, we predict the host range of multipartite viruses is greater than that of segmented viruses. Others have already noted that the number of genome segments is a predictor of virus host range, as plant viruses with 3–4 genome segments have broadest host ranges [52].

This straightforward hypothesis is not easy to test in practice. First, susceptibility of a host to a virus is not a binary characteristic: It depends on the conditions under which a host is exposed to a virus [53], as many variables such as the extent of exposure (i.e., dose), host immune state, and the presence of other viruses and microorganisms will affect whether infection occurs. Second, traditional host range tests will be limited by the number of candidate host-species tested, potentially underestimating host range and resulting in self-reinforcing biases about host compatibility (e.g., the assessment of additional candidate species for viruses with large known host range). We are therefore interested in measuring the realized host range for a large number of viruses in a manner that is not biased by the extent of host range testing.

We devised a less-biased approach for measuring the host range by using submissions from the NCBI Virus database, which contains high-throughput sequence (HTS) metadata from DNA and RNA viruses and records the host from which it is derived (see Methods Section 4.3). As the product of untargeted virus detection, each HTS-analyzed plant species contributes data on the naturally occurring viruses it may harbor, thereby capturing a large number of virus–host associations as they occur in the laboratory and the real-world. This approach will therefore render what is known as the observed host range [54,55]. There are a number of limitations to this approach, including the fact that the detection of viral sequences does not necessarily entail completion of the virus life cycle or even replication, and that ecological barriers that prevent the exposure of host species to viruses are not accounted for. Moreover, determining which plant species are being commonly sequenced (i.e., crops vs. wild plants) can also introduce bias, and sampling intensity and spatiotemporal coverage will vary per host. However, this HTS-based approach does avoid limitations inherent to testing individual virus–host combinations experimentally, as virus detection was untargeted.

For each plant virus species, hosts were identified to genus level by combining data from the NCBI Virus database [56] and ICTV virus metadata resource [57] (Table 1). For each virus species, we identified the number of unique host genera in which it had been identified. We performed our analysis on the host genus level, because we were interested in hosts that are highly diverging. We also included information on the number of observations for each virus species in our statistical analysis, as this number was variable. Consider that for each virus the host range cannot exceed the total number of observations, so it is important to correct for the number of observations. Therefore, we modeled the observed host range using a binomial error structure, with the number of different host species as successes and the total number of observations as trials. As each segment of a genome is a separate entry in the dataset, we corrected the number of observations for the multipartite and segmented viruses by dividing the number of segments observed by the mean number of segments for species that belong to a viral genus. The genus *Begomovirus* contains both monopartite and multipartite members, and was well represented in the dataset. We therefore classified this genus separately in the initial analysis.

Our analysis suggests that the genus-level host range is significantly narrower for the monopartite viruses than for all other groups considered (Table 1), providing support for our hypothesis that genome plasticity widens the host range of the segmented and multipartite viruses. By contrast, there is no support for the idea that multipartite viruses have a broader host range than segmented viruses: There were no significant differences between these groups, although the estimated host range was higher for the segmented viruses than the multipartite ones. Similarly, the *Begomovirus* genus, encompassing both monopartite and multipartite species, had an estimated host range that was significantly higher than the monopartite viruses and similar to the segmented and multipartite viruses. As there was considerable data for the begomoviruses, we performed a separate analysis comparing the host range of monopartite and multipartite species in this genus. For our dataset, the number of monopartite and multipartite species is similar, allowing a fair comparison. We found that the estimated host range was higher for the multipartite begomoviruses than for the monopartite ones, although the difference is not significant (Table 2).

Based on these analyses, we suggest that across all plant viruses, host range of segmented and multipartite viruses may be larger than that of monopartite viruses. As genome formula changes may not be pervasive among segmented viruses, these results also suggest that reassortment may play a more important role in host range expansion than genome formula change. For one virus family with many monopartite and multipartite species, however, the difference in host range was not statistically significant.

These results are congruent with previous work based on experimental host range determination [52], but we think they should be interpreted cautiously for a number of reasons. First, although we found significant differences in the overall analysis, the magnitude of the differences is not large. Second, as discussed before, although we corrected for the number of observations for each virus, there will still be biases introduced by the host species chosen and uneven sampling. Third, we analyzed host range at the virus-species level, but species demarcation criteria differ across viral genera and families [58]. Such differences may bias our analysis, particularly if levels of genetic divergence covary with host range. Fourth, the evolutionary history of a virus lineage could be a confounding factor, as many evolutionarily conserved traits, such as particle morphology, suppression of host immune defense, and mode of transmission, may influence host range. Phylogenetically independent contrasts between monopartite, segmented, and multipartite viruses would make these findings more robust. However, such analysis is not possible, as whether viruses are monopartite, segmented, or multipartite is typically a highly conserved trait, occurring uniformly within genera and often even within families. Finally, we stress that although a high adaptive potential may lead to a broader host range, a broad host range in itself is not necessarily beneficial or a direct consequence of selection for a particular virus trait.

## 3. Conclusions

Our analysis provides further confirmation of the long-held notion that multipartition imposes a cost on transmission [1,4,5,6,27]. By considering both the shape and position of the dose response, we show that the empirical cost of multipartition exceeds theoretical predictions, re-using experimental data of tobacco plants challenged with the tripartite virus AMV. Our re-analysis of data could not identify why the empirical cost of multipartition to transmission was higher than predicted. However, the most likely reason is the removal of barriers to virus spread within the host, as the original study noted the high levels of spread when one or two virus genome segments were constitutively expressed in plants [6]. The main insights from our analysis are the following: unequivocal establishment of a transmission cost for multipartite viruses cannot be done with the classical dose–response data which only considers the gradient or shape of the curve. A complete analysis also needs to consider how the position of curve shifts as the number of genome segments required for infection is modified. The limited data we could use for this purpose do support the notion that a multipartite organization is costly for between-host transmission.

One proposed mechanism for lowering the cost to within-host transmission of multipartite viruses is by gene-product sharing between cells during the course of infection [22]. We modelled the effect of gene-product sharing and showed that it can reduce the cost of multipartition. Our model predicts this reduction will occur at low MOIs, when the level of sharing of gene products is moderate, and when there is low sensitivity of virus yield to the genome formula. Recapitulating these three conditions more intuitively: gene-product sharing is likely to be helpful when genome segments are missing in many cells, when investment in this trait is limited because only a little sharing is done, and when the return on investment is large because it only takes a little sharing to achieve moderate levels of infection in other cells. Overall, there was only a small region of parameter space in which gene-product sharing was beneficial. This result suggests that gene-product sharing may not be common among multipartite viruses, unless this parameter space is representative for plant virus infections in the real world, which is largely unknown. Moreover, the parameter space in which gene-product sharing was beneficial does not overlap with the conditions under which models predict that multipartite viruses can outcompete monopartite ones [9]. Taken together, these results suggest that a virus population is unlikely to derive major benefits from both mechanisms simultaneously. This conclusion is intuitive: the genome formula framework requires virus replication to be sensitive to the stoichiometry of virus gene products, whereas the gene-product-sharing framework predicts the greatest benefits when the presence of a small amount of a missing gene product can lead to reasonable levels of replication in a cell missing the cogent segment. However, key parameters such as MOI will change during the course of infection, and, therefore, different mechanisms that lower the cost to transmission of multipartition could be important under different conditions. We present these model results to illustrate that reducing the cost of multipartition is complex, and that proposed mechanisms for reducing this cost to virus spread may incur additional costs.

Our confirmation of the cost of multipartition underlines the relevance of the classic question of why such a costly feature has evolved and is common among plant viruses [1,10,13,58]. Given the multifaceted diversity of multipartite viruses in terms of their evolutionary history, segment number, lifestyle, and virus-particle structure [1,10], we anticipate that the answer will be pluralistic. Moreover, it is often not easy to disentangle the causes and consequences of evolutionary change. For example, if within-host selection against cheaters could in itself drive the emergence of multipartition as suggested by theory [24,25], the emerging segmented genomes may still confer some benefits on the between-host level as an evolutionary by-product. These by-products, in turn, may be important for understanding why multipartite lineages have persisted instead of being outcompeted by lineages less prone to the evolution of cheating. Exploration of the putative benefits of multipartition, therefore, is relevant but needs to be framed carefully. The host range analysis we performed indeed suggests that both multipartite and segmented plant viruses have a broader host range than monopartite plant viruses, in agreement with earlier work using traditional host range metrics [52]. We therefore conclude that expanded host range, facilitated by the greater genomic plasticity of segmented and multipartite viruses, could be one of these by-products that is advantageous in some ecological contexts. The growing metagenomic data in which viruses can be readily identified allows many opportunities to study the role of re-assortment and genome-formula change in the adaptation of multipartite and segmented viruses to a wide range of host environments.

## 4. Materials and Methods

### 4.1. Re-Analysis of AMV Dose–Response Data

Here we provide a detailed description of how the data from experiments with AMV in plants expressing AMV genome segments [6] were reanalyzed. Briefly, in these experiments, transgenic plants expressing zero (*N. tabacum*), one (*P2*) or two (*P12*) AMV segments were inoculated with a wild-type AMV isolate. Two weeks post inoculation, leaves were sampled and AMV was detected by hybridization with a digoxigenin (DIG)-riboprobe for AMV RNA4, a subgenomic RNA of RNA3, which is not constitutively expressed by any of the plants.

As a starting point for considering the cost of multipartition, we consider the previously described infection model for a multipartite virus with *j* segments:$$I=\prod_{i=1}^{j}1-e^{-\rho_{i}n_{i}}$$
where *I* is the proportion of infected hosts, *ρ* is the probability of infection per virus particle, and *n* is the number of virus particles. The number of virus particles for each segment is known, as measured for purified virions from infected tobacco plants [18]. If we then assume *ρ_i_* is the same for all segments, the model only has a single free parameter. Conversely, if we estimate *ρ_i_* for each segment the model will have *j* free parameters. For the transgenic plants used in our dataset (*P1*, *P12*), the expression of an AMV RNA by the plant results in *ρ* = 1. Therefore, whereas the full model for 3 virus particle types is needed to describe AMV infection of wildtype plants $I_{wt}=\;\left(1-e^{-\rho_{1}n_{1}}\right)\left(1-e^{-\rho_{2}n_{2}}\right)\left(1-e^{-\rho_{3}n_{3}}\right)$, for *P2* plants: $I_{P2}=\;\left(1-e^{-\rho_{1}n_{1}}\right)\left(1-e^{-\rho_{3}n_{3}}\right)$ and for *P12* plants: $I_{P12}=\left(1-e^{-\rho_{3}n_{3}}\right)$.

We can make different assumptions on whether *ρ* depends on the virus-particle type and host plant, leading to four different models with a different number of free parameters (see also Figure 2). From first principles, we might expect that the probability of infection might be independent of the plant type, since we are modelling only the presence or absence of infection in the inoculated leaf. It is probably more difficult to make any predictions about whether the probability of infection depends on the virus-particle type, although previous work with this dataset suggests this is the case [6]. On the one hand, there are differences in the morphology of the virus particles that depend on the genome segment encapsidated [59]. On the other hand, all particles have the same capsid protein and similar physicochemical properties. One advantage of assuming that *ρ* is independent of virus-particle and plant types (i.e., Model 1), is that we only need values for the SGF to determine the cost of multipartition. In other words, the free parameter *ρ* can then shift the dose response in all three types of plants, but the shapes and relative positions of the responses are fixed. We consider Model 1 the null model, because of its parsimony (1 free parameter) and its property of fixing the cost of multipartition for a given SGF.

We then fitted all four models to the data using a stochastic hill-climbing algorithm to minimise the negative log likelihood (NLL). NLL was determined by assuming a binomial error structure such that for the *l*th plant type and *m*th dose:$$L\left(I_{l,m}|a,b\right)=\left(\begin{array}{c} a\\ b \end{array}\;\right){I_{l,m}}^{b}{\left(1-I_{l,m}\right)}^{a-b}$$

The corresponding NLL was then summed over all doses and plant types (i.e., each model was fit to all the experimental data). To obtain confidence intervals for parameter estimates, we fitted the model to 1000 bootstrapped datasets and then determined the 95% fiducial limits. The model fitting was done using a custom R script. We used the Akaike Information Criterion (AIC) for model selection. To estimate the 95% confidence band of the model predictions, we determined the predicted lower and upper 95% interval of the predicted responses over all bootstrapped datasets.

### 4.2. A Simulation Model to Explore Gene-Product Sharing in Multipartite Viruses

To explore the effects of gene-product sharing on virus fitness, and explore the relationship between the genome formula and gene-product sharing, we adapt a simulation model of multipartite virus genome-formula evolution [9]. This model assumes a bipartite virus with genome segments 1 and 2. Each genome segment produces a unique gene product that is needed for a successful cellular infection. If both gene products are present in a cell, the virus genome segments (1, 2 or both) in that cell will be replicated, incorporated into virus particles and can infect new cells in subsequent rounds of infection. There is a fixed number of cells (*c*) in each generation, there are discreet rounds of cellular replication with this fixed number of cells, and there is no spatial structure or differences in susceptibility between cells. To this model, we add the possibility of a donor cell in passage *t* sharing a proportion of the gene products it generates across all cells in passage *t* + 1. A key parameter in this model is *σ*^2^, a parameter which determines how changes in the ratio of the two viral gene products affect the virus’s ability to exploit a cell, either by producing virus particles or assembling gene products that can be shared with other cells.

At the start of each round of infection, cells are exposed to virus particles and gene products produced in the previous round of infection. In the original study, we fixed the cellular multiplicity of infection (*λ*), the total number of virus particles entering each cell. Here, we set a value for *λ*, but this number represents the maximum mean number of virus particles that infects a cell, and the realized mean can decrease due to a number of factors, as explained later on. There is a Poisson-distributed realization of the number of virus particles entering cells for each type (*λ*_1_ and *λ*_2_, i.e., virus particles containing genome segments 1 and 2, where *λ*_1_ + *λ*_2_ = *λ*).

Gene products are shared with units (*β*_1_ and *β*_2_) that are equivalent to the potential gene expression from an invading virus particle. However, we assume that these gene products are homogeneously distributed over all cells in each round of replication, such that the amount of shared gene product in each cell is e.g., $h_{1}=\frac{\beta_{1}}{c}$. Therefore, virus gene-product sharing is not subject to stochastic variation, as seen for the spread of virus particles.

The frequency of each genome segment in each cell then proceeds as before, e.g., $f_{1}=\frac{k_{1}}{\left(k_{1}+k_{2}\right)}$, where *f* is the within-cell frequency of a genome segment. We assume there is no within-cell competition between genome segments, and therefore *f*_1_ determines the relative frequency of the virus particle types produced by a cell. The ratio *r* of the virus gene products *g* for the two segments within a cell will determine the total virus resources of that cell: $r=\frac{g_{1}}{g_{2}}=\frac{\left(k_{1}+h_{1}\right)}{\left(k_{2}+h_{2}\right)}$. Under this model, a cell is infected and will produce virus resources if three conditions are met: $k_{1}+k_{2}>0$, $g_{1}>0$ and $g_{2}>0$. i.e., At least one genome segment must be present, whereas both gene products must be present, regardless of their origin (virus particles or gene-product sharing). The exponential kernel of the normal distribution is used to link viral proteins to virus resources generated, such that $\varphi \left(r\right)=exp\left[\frac{-{\left({log}_{10}\left(r\right)-µ\right)}^{2}}{2\sigma^{2}}\right]\;$. Here *µ* is the mean of the distribution, which is the value of ${log}_{10}\left(r\right)$ that results in the highest virus resources being generated, and *σ*^2^ is its variance, the parameter which determines how quickly virus resources drop as *r* moves away from the optimum *µ*. Here $\varphi \left(r\right)$ has a range between 0 and 1.

In a previous model, the function $\varphi \left(r\right)$ determine virus particle yield, but here we equate this with virus resources, because these resources generated can be dedicated to producing virus particles, producing viral gene products that are shared with cells in the next round of infection, or both. The proportion of these resources committed to sharing is *ρ*, with a range between 0 (only virus particles produced, no resources committed to gene-product sharing) to 1 (no virus particles produced, all resources committed to sharing). We also scale $\varphi \left(r\right)$ by λ to determine the virus particles produced, so that at the maximum virus particle yield possible, a constant MOI would be maintained over passages. Therefore, the contribution of each cell to virus particle production (*vp*) for segment 1 is ${vp}_{1}={\left(1-\rho \right)f}_{1}\lambda *\varphi \left(r\right)$, and the contribution of each cell to gene product (*gp*) sharing for segment 1 is ${gp}_{1}={\left(\rho \right)f}_{1}\lambda *\varphi \left(r\right)$. Unlike the previous model where MOI was fixed over rounds of passaging, suboptimal virus particle production over all cells will therefore lead to a reduction in MOI and possibly extinction of the virus. Note that the relative levels of production of the two gene products (*gp*_1_ and *gp*_2_) depends on the frequency of the two virus genome segments that entered the cell, analogous to virus particle production. We average over all cells in this round of infection, where for cells that are uninfected per definition ${vp}_{1}={vp}_{2}=0$ and ${gp}_{1}={gp}_{2}=0$.

We performed 10^4^ simulations per condition, as the individual simulations can give highly diverging results (e.g., due to the small number of cells *c* and low MOI *λ*). To measure the fitness of viruses in isolation, we calculated the aggregated virus accumulation over all cells in the simulation (*c* × *t_final_*), normalised by aggregated accumulation for a virus that does not share its gene products (*⍴* = 0), all other conditions being equal and using the mean values over all simulations. To explore model behaviour in a range of conditions, we varied model parameters *λ*, *σ*^2^, *c* and *ψ*, which determines the decimal log-transformed range in which *μ* can be sampled from a uniform distribution. I.e., when *ψ* = 0, *μ* is 1 and even gene expression always results in optimal exploitation of cells. When *ψ* = 2, *μ* is varied randomly resulting in values between 0.01 and 100. As plant viruses spread by cell-to-cell movement between cells with a rigid spatial structure, we thought that a very small population of cells was most appropriate (*c* = 5), although we also considered larger population sizes (*c* = 100). An overview of model parameters is given in Table 3, and an overview of model results for different parameter values is given in Appendix B.

### 4.3. Estimates of Host Range Using Virus-Sequence Metadata

To test whether there are systematic differences in host range depending on genome organisation, we performed a meta-analysis to determine host ranges across plant viruses with monopartite, segmented, and multipartite genomes (Figure 4). We investigated the viral host range using virus sequence metadata from the NCBI Virus portal database (https://www.ncbi.nlm.nih.gov/labs/virus/vssi/, accessed on 17 April 2023). The NCBI virus portal database provides an overview of the metadata associated with high throughput sequencing derived viral (meta)genomes. The NCBI Virus portal [56] was downloaded as a dataset consisting of metadata of whole and partial virus genome segments (*n* = 11,031,767). The metadata included host species, viral genus, viral species and viral family. We used the ICTV Virus Metadata Resource [57] (Version MSL37 released 2 December 2022) to add information on the Baltimore classification [60] and information on which kingdom is infected by each virus species. These two datasets were merged in R using *tidyverse* package *dplyr* [61,62]. We then selected only observations for which the viral genus was considered plant-infecting based on the ICTV VMR ‘host source’ associated with specific viral genera (i.e., host source == ‘Plants’). We used only plant viruses for the host range analyses to ensure a fair comparison, given that most of the known multipartite viruses infect plants. For the plant-infecting subset of virus–host observations, we assigned genome organisation to each observation by manual curation. The number of unique host genera detected per virus species were then tallied to give an estimate of the host range. The total number of observations per virus species was tallied and corrected for the number of segments. These numbers were then used in a model to test for differences in host range between the three different modes of genome organisation: monopartite, multipartite and segmented.

We developed a simple model to predict the genus-level observed host range (henceforth “host range”) conditional upon the number of observations of that virus. The observed host range is the number of unique host genera observed per virus species. Our model makes two main assumptions. First, host range within a group (i.e., monopartite, multipartite or segmented viruses) follows a zero-truncated geometric distribution over viruses, with a single free-parameter *θ* and a mean host range 1/*θ*. The observed host range is the number of host genera which the virus infects under the real-world conditions for which the data were collected. We choose a geometric distribution based on the intuition that most viruses will infect one or few host genera, but a small fraction of viruses will infect a large number of different genera. Second, we assume that the virus is evenly distributed over all host genera, to keep the model as simple as possible.

We can then generate a prediction for the observed host range conditional upon the number of observations of a virus based on an iterative approach. For each unique value *u* of the total number of observations of a virus, we first draw a value *v* of the host range from a geometric distribution with the *rgeom* function. Next, we randomly sample *u* integers with replacement from the set {1, 2, … *v*} with the *sample* function, and then determine the number of unique values *w*, which is the realized host range for this iteration. For each value of *u* we perform *z* iterations, and use Laplace’s law of succession to estimate the pseudo-likelihood of an observed host range of *x* genera:$$L\left(x|u\right)=-\frac{1+w}{2+z}.$$

Laplace’s law of succession is used to avoid likelihood values of 0 during the model fitting procedure, while this means that the pseudo-likelihood values depend on *z*. To estimate *θ* for a set of observations, we used a grid search to find the value of *θ* that minimizes the cumulative negative log likelihood (NLL). We bootstrapped virus species to estimate the 95% confidence intervals of *θ*. We first generated expectations of $L\left(x|u\right)$ for each value of *x*, *u* and *θ* in the dataset, and then used these expectations for model parameter estimation.

An overview of the model-fitting settings and results is given in Appendix C (Table A3). Although we varied the range for free parameter *θ*, the step size was always set to 0.01. The number of permutations used to generate the model predictions of the observed host range was varied over runs, but the number of bootstraps to determine the confidence interval was always set to 1000. After an initial run over the full range of *θ* with a low number of permutations, we restricted the range of *θ* but at the same time increased the number of permutations for more precise model parameter estimates. As we are working with pseudo-likelihoods and adjust predictions by Laplace’s law of succession, the number of permutations affects both the model parameter estimation and the lowest NLL. Therefore, although we are optimizing the model and obtaining more precise estimates of *θ*, in practice the NLL increases as the range of *θ* is restricted as more permutations are used in successive runs. One reason these effects occur can be that the very small or zero predicted likelihoods will affect the model parameter estimation more strongly, as with a larger number of iterations we gain more confidence that these values are indeed very small using this approach.

## Figures and Tables

**Figure 1 viruses-17-01275-f001:**
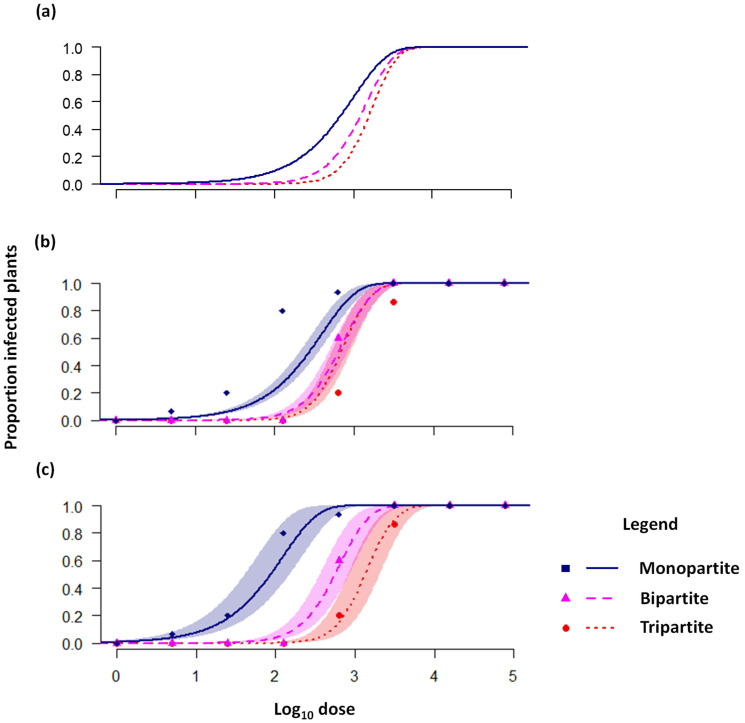
**An overview of predictions and data for the cost of multipartition.** Lines indicate model predictions, the shaded area is the 95% confidence band for the model prediction, and symbols (squares, triangles and dots) indicate empirical data. (**a**) Theoretical dose–response curves are shown for viruses with a different number of genome segments and balanced GF. As the number of segments increases, the dose response becomes steeper and shifts to the right. (**b**) Fitted Model 1 and experimental data from [6] are shown, where virus segments were made redundant by their constitutive expression in the host plant. For example, in the “monopartite” case, AMV RNA1 and RNA2 are expressed by the host plant and only RNA3 is required for infection. The dose–response predictions for the bipartite and tripartite viruses are different from those in panel a, both in terms of shape and relative position, because the AMV genome formula is not balanced. (**c**) Fitted model 3 and the same experimental data shown in (**b**) are shown. Models 2, 3, and 4 provide very similar predictions of dose response (Appendix A and Figure A2).

**Figure 2 viruses-17-01275-f002:**
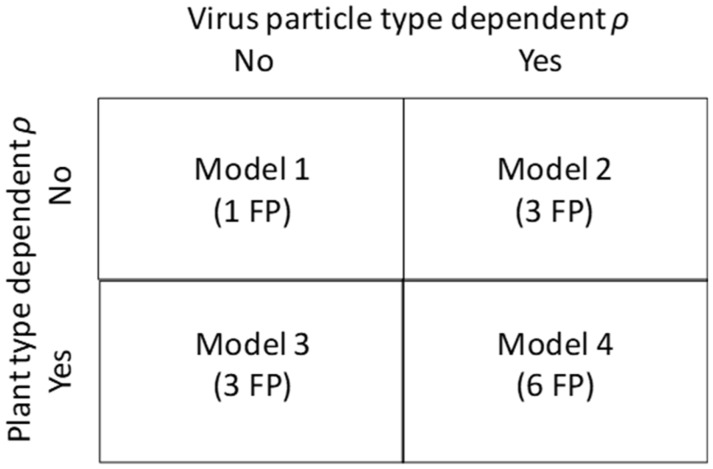
**Overview of the different models fitted here.** FP stands for free parameter, and *ρ* is the probability of infection per virus particle. Note the Model 4 only has 6 parameters (instead of 9) because when the transgenic plants express an AMV RNA, *ρ* is set to 1. Hence, for the *P2* and *P12* plants there are 2 and 1 free parameters, respectively.

**Figure 3 viruses-17-01275-f003:**
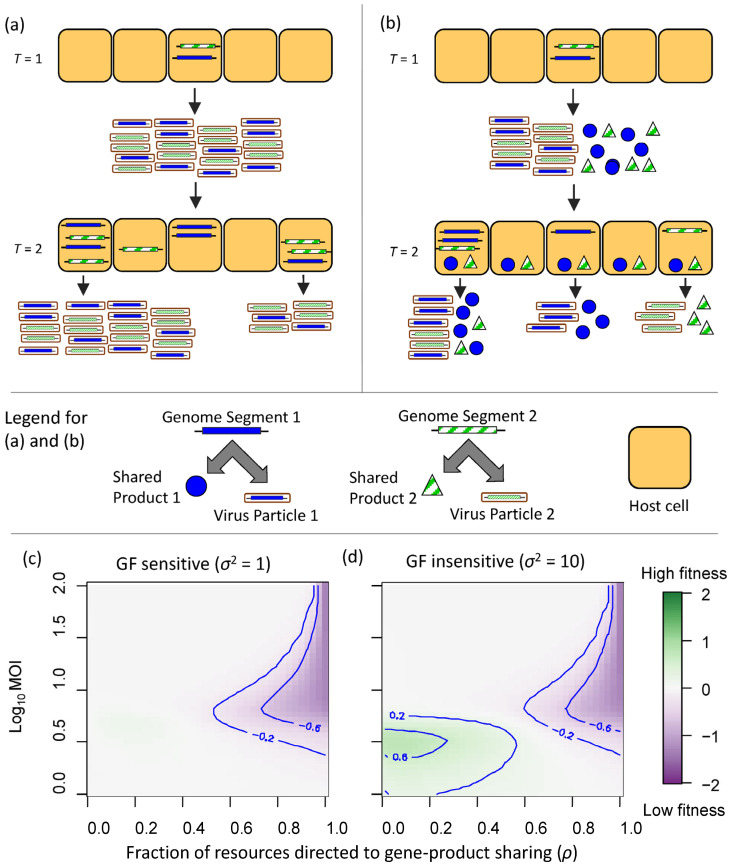
**Virus gene-product-sharing model**. Panels (**a**,**b**) illustrate the model, with a legend directly below the two panels. The blue and green bars illustrate the two different virus genome segments, and their cognate gene products are shown with blue circles and green triangles, respectively. When genome segments are packaged into virus particles, a rectangle is drawn around the segment. The large tan rectangles represent host cells. In the first round of infection (*T* = 1), a single cell is infected by virus particles in a 1:1 ratio for segments 1 and 2. In this example, we assume the production of virus resources is sensitive to changes in the proportion of gene products. A fraction *ρ* of the gene products in each infected cell will be shared uniformly with all other susceptible cells introduced in the next round of infection (*T* = 2). We only illustrate the gene products that are shared between cells. (**a**) There is no gene-product sharing and infection does not proceed in cells missing one of the two segments. When the genome formula deviates from 1:1, virus-particle production is lower. (**b**) Half of the gene products are shared (*ρ* = 0.5), and replication is supported in cells missing one segment. When the ratio of gene products deviates from 1:1 in a cell, the total virus resources generated are lower. When only one segment is present in a cell, only that segment can be replicated and its gene products produced. Panels (**c**,**d**) illustrate the effects of viral-gene-product sharing on the within-host fitness of a bipartite virus. The x-axis indicates the fraction of cellular resources used for gene-product sharing *ρ*, whereas the y-axis indicates the cellular multiplicity of infection (MOI). The colors indicate viral fitness compared to a virus that does not share its gene products, as determined by total virus particle production during a simulation of multiple rounds of infection. Contour lines have been included to highlight parameter space with high or low fitness. Model parameter *σ*^2^ determines how sensitive virus particle yield is to changes in the genome formula, with high values ($\sigma^{2}>1$) corresponding to low sensitivity. (**c**) When *σ*^2^ = 1, gene-product sharing was not beneficial. For lower values of *σ*^2^, gene-product sharing never leads to increased fitness (See Appendix B). (**d**) Gene-product sharing was beneficial in a small region of parameter space for the highest value of *σ*^2^, when virus particle production is insensitive to changes in the genome formula. For each condition, 10^4^ simulations were run with five cells per round of replication and 20 rounds of replication.

**Figure 4 viruses-17-01275-f004:**
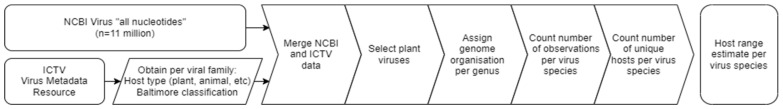
**Schematic overview of virus host range analysis**.

**Table 1 viruses-17-01275-t001:** Estimate of genus-level host range for different groups of viruses.

Group	N ^a^	Host Range ^b^ [95% CI ^c^]
Monopartite	869	3.03 [2.63–3.23]
Multipartite	196	4.17 [3.33–6.25]
Segmented	67	6.67 [4.00–12.50]
*Begomovirus*	413	4.17 [3.85–5.00]

^a^ The number of virus species per group. ^b^ The genus-level host range, estimated from 10^5^ permutations of the data. ^c^ Confidence interval, as determined by 1000 bootstraps of the virus species included in the model calibration.

**Table 2 viruses-17-01275-t002:** Estimate of genus-level host range for begomoviruses.

Group	N ^a^	Host Range ^b^ [95% CI ^c^]
Monopartite	245	3.33 [2.50–4.35]
Multipartite	168	3.85 [3.23–4.76]

^a^ The number of virus species per group. ^b^ The genus-level host range, estimated from 10^5^ permutations of the data. ^c^ Confidence interval, as determined by 1000 bootstraps of the virus species included in the model calibration.

**Table 3 viruses-17-01275-t003:** For the model of the effects of gene-product sharing on multipartite virus infection, for each model parameter we give the range of values used in the simulations, a brief explanation of the parameter, and miscellaneous comments for clarification.

Parameter	Value	Explanation and Comments
*λ*	$10^{0},\;10^{0.1},\;\;\ldots ,\;10^{2}$	Cellular multiplicity of infection; maximum value that can decrease under suboptimal conditions.
*c*	5, 100	Number of cells per round of infection. Unlike the previous model, *c* is not adjusted to maintain the same number of effectively infected cells per round of infection.
*σ* ^2^	0.01, 0.1, 1, 10	Variance of the normal probability function used to link virus gene products to the total production of viral resources
*t_final_*	20	Number of rounds of infection
*ρ*	0, 0.05, 0.1, …, 1	Proportion of virus resources dedicated to gene-product sharing, the remaining fraction is used for virus particles.
*ψ*	0, 2	Determines the range of values from which values of μ can be drawn from a uniform distribution, from 0−ψ to 0+ψ. When ψ = 0, μ is always 0.

## Data Availability

This study only uses publicly available data, although we have made the input data used in our scripts available to readers for convenience. All data and code used is available here: https://doi.org/10.5281/zenodo.17106488, accessed on 12 September 2025.

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
