# Peer review of "Living Together Apart: Quantitative Perspectives on the Costs and Benefits of a Multipartite Genome Organization in Viruses"

_viruses, 2025, doi:10.3390/v17091275_

Round 1
Reviewer 1 Report
Comments and Suggestions for Authors
This study uses a combination of mathematical / computational modeling, reanalyses of experimental data, and surveys of viral genetic databases to evaluate some of the fitness costs and benefits associated with why viruses, particularly in plants, might exhibit multipartite genomes. I found the study ambitious and the authors do a decent job bringing a surprising amount of cohesion in presenting potentially disparate analytic approaches.
The flip side is this does expose them to some criticism from each of these facets. Still, methodologically the authors results appear generally internally consistent. Hence, whilst I may not be as thoroughly convinced about some of the modeling and data analysis decisions the authors make (described below), on the whole I think this study represents a useful contribution. I feel there are some missed opportunities, but won't insist on the authors pursuing all of these, given that this is already a quite dense, sprawling manuscript. So these comments are probably more for points that could be better flushed out and giving the readers a better sense of the extent of the advance presented here.
Broader comments:
1. There's a question if host range per se is adaptive. As I'm sure the authors are well aware, expanded host ranges aren't always indicative of a pathogen's trait being adaptive (or as being a selective pressure for the trait). They sort of sidestep this somewhat circular argument in generating their hypothesis about why we see segmented viruses (it allows for broader host ranges -> which can show being multipartite is adaptive -> which seems to me to suggest that this presumes broader host ranges are adaptive). I realize the authors aren't the first to make this sort of argument, but there has to be a bit more nuance in its presentation. There's some of this in the conclusion, but I'm not sure that quite gets at the heart of the problem.
2. On some level a model where you have two competitors compete within the same system (even without coinfection) struck me as maybe more informative of whether being multipartite outcompetes the monopartite alternative (a point the authors allude to in presenting their results, based on the modeling study they build this study upon). My biggest unhappiness is that proxy measures of fitness and fitness over time can be a notoriously slippery concept to pin down and define (e.g., Metz et al. 1992), and total reproductive output has well attested limitations. I'm not entirely convinced by the authors reasons for setting aside the serious issue of exploitation around line 370, since if the goal is to explain the strategy's evolutionary viability this seems to be a major omission. Although I won't insist on the authors redoing their study from scratch to allow for head-to-head competition, given that these two types of strategies do co-exist in nature (e.g., lines 511-513), they could be more upfront about this limitation of their analysis.
Fig. 5, line 426 and lines 668-669 -> does counting the number of observations per virus species (step 4 after the databases are merged) really help? The authors claim around line 426 that they use it to weigh the diversity measure of host range. I'm not really sure this works, however. For instance the sampling effort is far from random in these kinds of databases (at least for now). Then the material on lines 685-690 suggests to me this might be to account for sampling issues or the problem of bias mentioned around line 400, but I'm not entirely sure this reasoning made sense to me either.
More specific comments:
Line 94 cheats -> cheaters. There are also parallels/echoes here to the transposable elements literature that the authors could consider citing.
Line 158-> "whereas the position of the curve is not defined or considered" I'm not sure I understand what the authors mean by this. The next sentence suggests otherwise viz. the authors claim; if it's a sigmoidal curve the position should vary along (and be defined by) the 50% mark/inflection point and its steepness.
Figure 3b-c -> Horizontal /vertical confidence intervals perhaps based on their fiducial limits in the fitted curves could help, especially as the data points at intermediate doses are sparse.
Line 227 -> Typo
The point Line 230 seems to just repeat a point made earlier.
Lines 256-258: not to mention adding a third dimension (vector type) to plant/virus type effects!
Line 312 (and elsewhere) -> "in a small region of "
Lines 314-> are these conditions more plausible for plants than, say, animals or phages in biofilms?
Fig 3c. Results would be easier to understand if parameter definitions (even if abbreviated) rather than symbols were used.
Lines c. 410~420 -> A point for discussion - I wonder if eDNA can solve a lot of these issues, particularly in relatively controlled settings like, say, horticulture where host community composition can be varied.
Line 485-> Related to my comment above, I guess by "position" the authors mean the 50% point of the sigmoidal curve? If so, if memory serves this is maybe better described as the inflection point.
Lines 505-507 -> I agree the model results indicate mutual exclusivity of parameter space. But I don't think, logically, these processes are mutually exclusive the way the authors describe or rephrase the underlying intuition. Put somewhat more explicitly, if clumsily, the use of stoichiometries does not imply the lack of qualitative regime shifts (and vice versa).
Methods:
A very brief characterization of how the dose response data were collected for the plants would help general readers, even if this information is available in other studies.
Line 565 (a b) -> presumably this is the combination; typically C(a,b), C^a_b or a vertical representation of a over b is used for notation.
606 typo Ie: It might be useful to remind reviewers here that k_i is not the same as k used for plant type on line 2. I'd suggest a different parameter symbol.
610: it's unclear if just r is log transformed (in which case the definition of mu I think is a little off) or if the authors intended to log transform r-mu. Also, I'm not sure the scaling by 1/sqrt(2ps^2) is needed since thinking of resources or even yield as a quantity between 0 and 1 seems strange (especially since it gets scaled by phi(1) later anyway and esp. in light of their qualitative conclusions starting in line 314). It's probably not wrong, just seems a bit eccentric / bloated.
628: average, not mean, right?
Tabel 3: presumably dots are ranges - dash is more conventional
Lines 637-638: I guess it's fine to use very small cell populations - this might be reasonable in plants e.g., where spatial structuring of cells is quite rigid so most infection dynamics happen on a very local scale (although this is perhaps unlikely in light of the point on lines 481-483). Still, presumably you'd get more consistent parameter signals when the host population size is larger.
Line 664: Well, at least those studied so far; I wouldn't be surprised if there was a reporting bias towards plants over, say, fungi
Reviewer 2 Report
Comments and Suggestions for Authors
Dear authors,
I found the research to be both fascinating and highly significant for understanding the evolutionary dynamics of multipartite viruses. Your multi-pronged approach to investigating the costs and benefits of this genomic organization is comprehensive, and the findings, particularly those related to host range, are a notable contribution to the literature.
I am pleased to recommend the manuscript for acceptance with minor revisions. My main suggestion is a request for a small addition to the Discussion section.
While the manuscript effectively explores the broader concepts of multipartite viruses, it seems to focus heavily on Ilarviruses as a primary example. It would be beneficial for the reader if the authors could briefly discuss in the Discussion why the manuscript focuses on this specific group, and perhaps touch upon how findings might compare or contrast with other multipartite or segmented viruses that are not extensively covered. This would help provide a more holistic context for the research and strengthen the overall discussion.
This is a minor point, and I believe addressing it would further enhance the clarity and impact of your excellent work.
Thank you once again for your excellent manuscript
Round 2
Reviewer 1 Report
Comments and Suggestions for Authors
The authors have satisfied my concerns.
Author Response
We thank the reviewer for looking at the manuscript again. We are greatly indebted for the comments they provided us with.